# Cyclic Load Impact Assessment of Long-Term Properties in Compression to Steel and Polyvinyl Alcohol Fibre-Reinforced Geopolymer Composites

**DOI:** 10.3390/ma16186128

**Published:** 2023-09-08

**Authors:** Rihards Gailitis, Andina Sprince, Michał Łach, Pavels Gavrilovs, Leonids Pakrastins

**Affiliations:** 1Faculty of Civil Engineering, Riga Technical University, LV-1048 Riga, Latvia; andina.sprince@rtu.lv (A.S.); leonids.pakrastins@rtu.lv (L.P.); 2Department of Materials Engineering, Faculty of Material Engineering and Physics, Cracow University of Technology, Jana Pawła II 37, 31-864 Cracow, Poland; 3Faculty of Mechanical Engineering, Transport and Aeronautics, Riga Technical University, LV-1048 Riga, Latvia; pavels.gavrilovs@rtu.lv

**Keywords:** fly-ash-based geopolymer composite, long-term properties under cyclic load, fibre-reinforced geopolymer

## Abstract

This study investigates the cyclic load application impact on fly-ash-based geopolymer composites that are reinforced with a low amount of fibre reinforcement. For reinforcement purposes, polyvinyl alcohol and steel fibres are used. For testing purposes, four geopolymer composite mixes were made, three of which had fibre reinforcement. Simultaneously, specimens were tested for shrinkage, static-load-induced creep, and cyclic-load-induced creep. For static and cyclic creep testing, specimens were loaded with 20% of their strength. For cyclic creep testing, load application and release cycles were seven days long. When each cycle was introduced, the load was applied in steps. Necessary load application or unloading lasted for 5 min and consisted of four steps, each 25% of the necessary load. From the long-term static and cyclic creep tests, it was seen that only the plain specimens showed that static creep strains are within cyclic creep strains. For all the other specimens, the static load was higher than the cyclic-load-induced creep amplitude. Also, 1% polyvinyl alcohol fibre-reinforced specimens showed the most elastic characteristics under cyclic load, and 1% steel fibre-reinforced specimens appeared to be the most resistant to the cyclic load introduction.

## 1. Introduction

As is known, unreinforced cementitious materials exhibit high brittleness and low tensile strength. Therefore, they do not have favourable mechanical properties in case of dynamic loading, such as earthquakes, explosions, impacts, and others [1]. Also, cyclic loading is not desirable for the structure of offshore platforms and railway infrastructure [2]. Furthermore, it has been shown and calculated that the existing reinforcement in concrete bridges would not be as restrictive to long-term strains, as it was anticipated in the design stage, but according to currently available creep models, bridges would exceed these strains significantly [3]. Many scientists [4], including Bažant [5], concluded that the superposition of static creep loading and cyclic loading increases long-term strains on concrete. If these strains are underestimated, it can result in significant damage and concerns safety aspects.

Static loading caused by total strain (*ε_tot_*_, *cr*_) is divided into elastic strain (*ε_el_*) and time-dependent creep strain partition (*ε_υ_*_,*cr*_), as shown in Figure 1b. Creep strain, as such, can be partialized into viscoelastic (*ε_υ_*_,*el*_) and viscoplastic (*ε_υ_*_,*pl*_) parts. In cases where the applied stress level exceeds 40% of the ultimate stress level, then additional plastic strains are caused due to structural changes in the microstructure. In this case, the plastic strain caused by a structural change in the microstructure is considered as a part of the viscoplastic strain (*ε_υ_*_,*pl*_) part. In cases of high-stress applications, the strain development can be divided into three groups:In the first stage, strains increase greatly;In the second stage, strains develop more moderately;In the third stage, strains again increase rapidly until failure due to high-stress application [3].

Strain development throughout data recording is mainly connected with microcrack development, and its amount increases throughout testing time [3].

The cyclic load applications (Figure 1c) caused by total strain can be partialized into elastic strain (*ε_tot_*_, *cycl*_) partition and a time-dependent, viscous strain (*ε_υ_*) partition (Figure 1d). Additionally, a load-cycle-dependent plastic strain is introduced due to damage [6]. The viscous strain part (*ε_υ_*) can be divided into a viscoelastic (*ε_υ_*_,*el*_) part and a viscoplastic strain part (*ε_υ_*_,*pl*_). The strain growth at a minimum and maximum stress peak (corresponding to *ε_tot_*_,*max*_ and *ε_tot_*_,*min*_) is usually investigated with the cyclic strain analysis. The maximum strains can be divided into elastic (*ε_el_*) and viscous (*ε_υ_*) parts. The strain development for *ε_tot_*_,*max*,_ and *ε_tot_*_,*min*_ can be divided into three parts:Disproportionate strain increase. This is due to the increased growth of microcracks and consequent plastic settlement during the first loading cycles;Linear strain increase. In this part, the microcrack development speed is stable with a diffuse character;Disproportionate strain increases up to specimen failure. It is only reached in high-stress-level cases. It happens due to the unstable growth of the microcrack net which connects and develops earlier, caused by microcracks [3].

The most widely used construction material for such civil infrastructure as tunnels, bridges, and roads, as well as buildings, is Portland cement (PC). For these structures, the most preferable and widely used reinforcement material is steel. Due to the fact that the population keeps on increasing, the need and demand for infrastructure and civil building will keep increasing; therefore, it is not likely that the demand for Portland cement and steel will stabilise. On the contrary, most likely, the demand will increase, and, consequently, the CO_2_ emissions will increase as well [7]. It has been estimated that cement production alone contributes 5% of global CO_2_ emissions [8]. According to various assessments, replacing Portland cement as a binder with a geopolymer binder brings CO_2_ emissions from a moderate 9% to 64% [9,10]. Geopolymer composites are considered a sustainable alternative to PC-based composites, mainly because they utilise industrial waste materials as binders, such as fly ash and various slags. Geopolymer has great resistance to fire, acid, and sulphate attacks, and if the design is right, it can have high compressive strength [7].

It has been claimed that fibre reinforcement for geopolymer composites would bridge the cracks, reduce the stress intensity at the ends of cracks, and prevent them from growing. Fibre reinforcement improves the load-bearing capacity, abrasion resistance, and fracture toughness of the specimen as well as the mechanical and ductility properties of the geopolymer composites. The most widely used fibres are steel, PVA, basalt fibres, and some natural fibres as well [11,12]. The main point of fibre additions to geopolymer composites is to increase the tensile strength and flexural strength. Additionally, fibres will increase the durability and stability of the composite under impact loads as well as enhance its plastic capacity prior to failure [13].

Steel fibre-reinforcement usage as a replacement reinforcement for steel bars has increased in the past decades. The main reasons for such changes are the relatively easy usage of the fibre reinforcement and the increased corrosion resistance of the concrete that has been reinforced with steel fibres. In steel fibre-reinforced concrete, the fibres are introduced in low amounts to increase toughness and ductility [2]. These fibres have high mechanical strength and flexibility and are easily available [14,15].

Polyvinyl alcohol fibres, unlike polypropylene (PP) or polyethylene terephthalate (PET) fibres, have a high tensile strength and modulus of elasticity. They also have stronger bonding characteristics with the cementitious matrix due to hydroxyl groups in their molecular [16] chains [17,18]. These flexible fibres exhibit a deflection hardening behaviour, creating a bridging effect over stress-induced fractures, thus increasing a material’s strength [19]. Furthermore, drying shrinkage is reduced due to this bridging effect, and it has been reported that in the case of fire, PVA fibres melt, creating channels for water vapor to expand, therefore reducing the vapor pressure within the specimen and possible cracking destruction of it. The main properties of the steel and PVA fibres used in concrete composites are shown in Table 1.

It has been shown that compression at low load levels causes the strengthening of concrete specimens. Under high levels of compression, the non-linear mechanical behaviour of concrete emerges [22].

This research aims to see if static loading in compression causes lower amounts of creep strains than cyclic loading and if fibre reinforcement helps to reduce the static and cyclic loading impact on the creep strains of plain and fibre-reinforced geopolymer composites. This article’s main point is to evaluate plain and fibre-reinforced geopolymer composites’ ability to resist long-term cyclic loading and to determine whether the creep strains for these composites is within static creep strains, therefore showing that these types of composites could be suitable for such structures as bridge supports or piles.

## 2. Materials and Methods

The matrix of the geopolymer composite specimens was based on fly ash. Fly ash was mixed with sand in a proportion of 1:1. As an activator, 10 mol NaOH and Na_2_O + SiO_2_ solution were used, which were added to the sand and fly ash mix after the sand and fly ash were mixed together in a dry state. The dry mix and geopolymer mix compositions are presented in Table 2, and the fly ash chemical composition is in Table 3.

For testing purposes, four geopolymer mixes of equal specimen amounts were prepared. Four reinforced specimens, 1% polyvinyl alcohol (PVA), 1% steel, 0.5% PVA, and 0.5% steel fibres were used. Used fibre parameters are shown in Table 4.

Also, specimens without reinforcement were prepared. Precise preparation procedure and fly ash chemical composition can be found in [24,25,26]. In Figure 2, plain and fibre-reinforced geopolymer composite mixtures are shown.

Specimen size and, consequently, long-term tests were performed based on RILEM TC 107-CSP recommendations [27]. Specimens were cylinders with dimensions of Ø46 × 190 mm or approximately 1:4 diameter-to-height ratio. For stain gauge attachment, six aluminium plates (10 × 15 mm) were glued in pairs to each specimen. Afterward, strain gauges were attached to those plates. The same preparations were conducted for the shrinkage specimens. Long-term strain tests (shrinkage, static creep, cyclic creep) were performed in laboratory conditions with controlled atmosphere conditions: temperature 24 ± 1 °C and relative humidity 30% ± 3%. Specimen placement in test stands, as well as shrinkage specimen placement, is shown in Figure 3.

The shrinkage and static creep strain readings were monitored every day for the first two weeks, after which they were monitored every other day. A constant load was applied throughout the whole static creep testing period. The specimens were loaded with a load equivalent to 20% of the ultimate compressive strength, which was determined in compressive strength tests prior to long-term tests. The readings of the cyclic creep strains were monitored every day, especially on the day when specimens were loaded or the load was taken off. The readings were taken prior to the load application or unloading, and every hour until within a spec of one hour, the strain did not change. The cyclic creep specimens were gradually loaded up to the same stress level as the static creep specimens by steps of 25% of the necessary load in a short period (within 5 min). Cyclic loads were applied and unloaded every 7th day.

## 3. Results and Discussion

### 3.1. Compressive Strength of Plain and Fibre-Reinforced Geopolymer Composites

The compressive strength of the specimens was determined before the creep tests at the age of 28 days. The compressive strength was determined to be a total of 16 specimens (4 from each geopolymer composite composition) that were the same shape as the creep specimens. The compressive strength results are represented in Figure 2.

The necessary load that has to be applied to creep test stands was calculated from the average compressive strength values shown in Figure 2. Also, if the Figure 4 results are looked at together with the claims made by Moradikhou [28], that the compressive strength should slightly improve if fibre reinforcement is used, it is visible that in this case, all of the fibre-reinforced composites show lower compressive strength values than plain geopolymer mortar specimens. It might seem that the used fibre amount is not sufficient, but as Ravinder et al. has shown, when the used fibre dosage is increased from 0% to 0.3%, the compressive strength should increase by 6% [29]. From Figure 2, it is clear that none of the used reinforcement amounts improve compressive strength capacity. The compressive strength, in contrast to plain specimens, decreased by 20.22%, 17.66%, and 6.00% for 1% PVA fibre-, 1% steel fibre-, and 0.5% PVA and 0.5% steel fibre-reinforced geopolymer composite specimens, respectively. It has been reported that an increase in compressive strength is possible up to a certain amount of the specific fibres. When this amount is exceeded, compressive strength gradually reduces. Authors have found out that for high-strength concrete fibre usage, there is an up to 0.60% by volume increased compressive strength. When this amount is exceeded, compressive strength dropped significantly to the compressive strength of plain high-strength concrete [30,31]. Furthermore, as has been claimed by Ozel [32], the addition of polymer fibre caused a reduction in compressive strength due to an increase in porosity and a greater number of weak areas in the specimens. In their case, steel fibre inclusion increased the compressive strength, which can mainly be attributed to the size of fibres, as they were 6 mm, while in this study, the steel fibres were 12 mm long. In this case, it is clear to see that because of the relatively large size of the PVA and steel fibres, the specific surface area of the fibres in the specimen is insufficient to make any improvement in compressive strength values. To improve compressive strength values, it would be necessary to increase the fibre amount to 3%.

### 3.2. Long-Term Properties of Plain and Fibre-Reinforced Geopolymer Composites under Static and Cyclic Loading

The static and cyclic creep, as well as shrinkage tests, began right after the initial compressive strength test. The static creep, cyclic creep, and shrinkage tests lasted for 105 days. The static creep and cyclic creep test active phases were 91 days with a 14-day relaxation stage or, in other words, on the 91st day, specimens were unloaded (for cyclic creep specimens, that was the last unloading), and they were kept unloaded for 14 days (2 weeks), as shown in Figure 5. The measurement standard deviation and coefficient of variation values for static creep, cyclic creep, and shrinkage tests are summarised in Table 5.

As becomes apparent from Table 5, the standard deviation values are low. Furthermore, all of the coefficient of variation values are less than one, which means that variation in the measurement values is low between the strain gauges, and there is no error regarding strain gauge slippage or similar dysfunction in the measuring apparatus. As was expected, the coefficient of variation for the cyclic creep measurements varied in greater amplitude (0.4800 for cyclic creep, 0.2063 for static creep, and 0.4089 for shrinkage measurements) than for all other measured strains.

Figure 5 shows that the static creep strains are significantly higher than cyclic creep strains. The exception is the plain geopolymer composite. On average, static creep strains are 21.70%, 37.69%, and 27.64% larger than cyclic creep strains for 1% PVA fibre-, 1% steel fibre-, and 0.5% PVA and 0.5% steel fibre-reinforced geopolymer specimens. For plain geopolymer composites, the cyclic creep strains are, on average, 21.12% larger than the static creep strains. A similar correlation between previously stated cyclic creep strains and static creep strains is visible when the specimens are unloaded. For all of the fibre-reinforced specimens, static creep strains hold a significantly higher amount of plastic strains than those subjected to cyclic load strains. The exception is the plain geopolymer composite specimens showing lower plastic strains for static creep curves.

The static and cyclic creep strains are shown in Figure 6.

Yet, it is not entirely conclusive from Figure 4, Figure 5 and Figure 6 which one of the tested geopolymer composites shows the lowest possibility to creep under static and cyclic load. Therefore, the specific creep is shown in Figure 7.

From Figure 7, it is apparent that the largest specific creep under static load is for the 1% steel fibre-reinforced geopolymer composite, which is followed by almost identical specific creep under static load values of 1% PVA fibre-reinforced and 0.5% PVA and 0.5% steel fibre-reinforced geopolymer composites; these are, on average, 11.41% and 12.89% lower. The specific creep value under static loading is surprisingly low for the plain geopolymer composite, and, on average, it is 32.40% lower than the specific creep value under a static load for the 1% steel fibre-reinforced geopolymer composite. In other words, the static and cyclic creep strains, if put on equal terms from the point of applied load, show that plain geopolymer specimens under static load are the least likely to creep from all of the tested geopolymer specimens. It further shows that this amount of fibre addition to the geopolymer composites makes them weaker. In the case of cyclic-load-specific creep values, it is the other way around, as the plain geopolymer composites exhibit the highest specific creep values. It means that under cyclic load, plain geopolymer deteriorates its structure, while for fibre-reinforced specimens, it stabilises, and the specific creep values even reduce from the 56th day onwards.

The creep modulus is one of the important parameters regarding polymer material properties. It can be defined as the instantaneous elastic modulus of the specific material that varies throughout time. As the creep modulus is inversely proportional to the creep, it shows material creep strain capacity (in cases of constant strain) and material elastic behaviour. According to ASTM D 2990-1 [33,34], the creep modulus can be calculated using this equation:E_cr_ = σ_c_/ε_c_(1)
where:

E_c—_Creep modulus;

σ_c—_Constant applied stress;

ε_c—_Creep strain at a specific time.

As can be seen from Figure 8, in the case of cyclic creep (Figure 8a), all specimens except plain geopolymer composites exhibit creep modulus increases and decreases throughout the testing time, while for static creep specimens, the creep modulus (Figure 8b) does exhibit saturation through the testing time that begins from the 28th day of testing and continues until the release of applied stress. Similar behaviour is exhibited in plain geopolymer composite specimens in cyclic load applications. It has to be mentioned that the creep modulus for the plain geopolymer under cyclic loads is more than 30% lower than that under static loading, therefore showing a lower creep strain capacity and leading to a conclusion of microstructure deterioration. Also, from Figure 8a, the creep modulus of the 1% PVA and 1% ST specimens appear close, while specimens with 0.5% PVA and 0.5% ST fibres have around 16.2% less creep modulus. The cyclic creep modulus values of fibre-reinforced geopolymer composites are close to plain geopolymer creep modulus values under static load cases. While values are close, the characteristics of the creep modulus are totally different, where the plain geopolymer under a static load has a saturation effect of the creep modulus under cyclic load fibre-reinforced geopolymer specimens do not show any saturation of the creep modulus. It further indicates the preferable elastic behaviour of the fibre-reinforced geopolymer composites under cyclic loads.

The cyclic creep strain and shrinkage strain relation is shown in Figure 9.

As is visible in Figure 9, shrinkage for plain and 1% PVA fibre-reinforced geopolymer composites is almost identical, as it is for 1% steel fibre, 0.5% PVA fibre-, and 0.5% steel fibre-reinforced composites. The shrinkage strain development for plain and 1% PVA fibre-reinforced geopolymer composites is significantly more rapid than for 1% steel fibre and 0.5% PVA fibre, and 0.5% steel fibre-reinforced geopolymer composites. For the first 64 days of the testing, the 1% steel fibre and 0.5% PVA fibre- and the 0.5% steel fibre-reinforced geopolymer composites show, on average, 30.34% shrinkage strains of the plain and 1% PVA fibre-reinforced geopolymer composite shrinkage strains. After the 64th day of the testing, plain and 1% PVA fibre-reinforced composite specimens show, on average, 38.31% lower shrinkage strains than the 1% steel fibre and 0.5% PVA fibre- and the 0.5% steel fibre-reinforced geopolymer composites.

To further evaluate cyclic creep strain differences from static creep strains, in Figure 10 the cyclic creep strain maximum top peak and minimum bottom peak points are made into separate curves and are represented together with the static creep strain curve surface area in the graph.

In the graphs of Figure 10, it is apparent that only the geopolymer composite that has static creep strains within the area between the curves made from cyclic creep-strain curve maximum top peak and minimum bottom peak points is plain geopolymer composite (Figure 10a). Static creep strains for all other geopolymer composites step out of the area between the previously mentioned maximum and minimum curves. This further shows that fibre inclusion in the geopolymer matrix does reduce the ability of plastic strain development. The most significant improvement is to the 1% steel fibre, which reduces plastic creep to a peak of 37.19% on average and is followed by 0.5% PVA and 0.5% ST and 1% PVA fibre-reinforced specimens with cyclic creep top peak reductions of 33.06% and 24.40%, respectively. For plain specimens, cyclic creep top peaks are 21.98% higher than static creep values.

Also, as shown in Figure 10c, the 1% steel fibre-reinforced specimen, the difference between the cyclic creep top and bottom peaks is lower than all the rest of the tested geopolymer composites. For the plain specimens, the difference or the amplitude of the top and bottom peaks of cyclic creep is 14.88% higher than that of 1% steel fibre-reinforced specimens. For 0.5% PVA and 0.5% ST and 1% PVA fibre-reinforced specimens, this amplitude is 51.02% and 62.35% higher than 1% steel fibre-reinforced specimens, respectively. For 1% steel fibre-reinforced specimens, the amplitude of the top and bottom peaks of cyclic creep is 0.00025689 mm/mm on average.

From the Figure 10 curves, it is clear that fibre inclusion into the geopolymer matrix makes the material act in compression under cyclic load application more elastically than without fibre addition. Furthermore, the addition of PVA fibre makes the material more elastic than the addition of steel fibre. This can be drawn from the top and bottom amplitudes of cyclic creep, where, for 1% PVA fibre-reinforced specimens, the bottom cyclic creep values are, on average, 45.45%, 65.26%, and 80.34% lower than those for 0.5% PVA and 0.5% ST, 1% ST, and plain geopolymer-composite cyclic creep bottom values. Furthermore, 1% PVA fibre-reinforced geopolymer-composite cyclic creep top peaks are within a 3% gap of the 1%ST and 0.5% PVA and 0.5% ST fibre-reinforced specimen and only 35.19% lower than the plain specimen cyclic creep top values. Still, the cyclic creep strain amplitude for 1% PVA fibre-reinforced specimens is the largest of all the tested specimens, 0.000682 mm/mm, respectively.

## 4. Conclusions

The long-term property tests under static and cyclic loads were conducted for a total of 105 days. For 91 days, the specimens were intended for long-term cyclic-load-induced creep measurements. The main conclusions are:Under static loading, fibre inclusion does not always reduce long-term properties. For creep reduction, the most effective was 1% PVA fibre usage, which still was 4.36% higher than plain specimen creep strains under static loading. By using 0.5% PVA and 0.5% St and 1% ST fibre reinforcement, the creep strains were 17.06% and 18.05% higher than those using plain specimens.Under cyclic long-term load application, the most resistant or those with the lowest cyclic creep values and strain amplitudes were specimens with 1% steel fibre reinforcement. Plain specimens had 14.88% higher amplitude; 0.5% PVA and 0.5% ST and 1% PVA fibre-reinforced specimens had a 51.02% and 62.35% higher cyclic creep amplitude range than 1% ST fibre-reinforced specimens. The cyclic creep, on average, was 3.03%, 5.03%, and 14.88% higher than that for the 1% PVA, 0.5% PVA, and 0.5% ST fibre-reinforced specimens and plain specimens, respectively.The most elastic under long-term cyclic load application were 1% PVA fibre-reinforced specimens that, on average, had 23.14%, 55.77%, and 62.35% higher cyclic creep amplitude than 0.5% PVA and 0.5% ST, plain specimen, and 1% ST fibre-reinforced specimen cyclic creep amplitudes. They also showed the lowest bottom values for cyclic creep of 0.0021 mm/mm, which was 45.45%, 65.26%, and 80.34% lower than the 0.5% PVA and 0.5% ST, 1% ST fibre-reinforced, and plain geopolymer specimen bottom cyclic creep values.All in all, this study shows that compression fibre-reinforced geopolymers show more reduced creep strains than under static loading. With the addition of polymer fibres, more elastic behaviour and greater strain amplitude can be achieved, while with steel fibres, the strain amplitude can be reduced for geopolymer composites under cyclic loads. This further leads to the conclusion that fibre-reinforced composites can be used for structures such as bridge foundations and supports subjected to cyclic loads.

## Figures and Tables

**Figure 1 materials-16-06128-f001:**
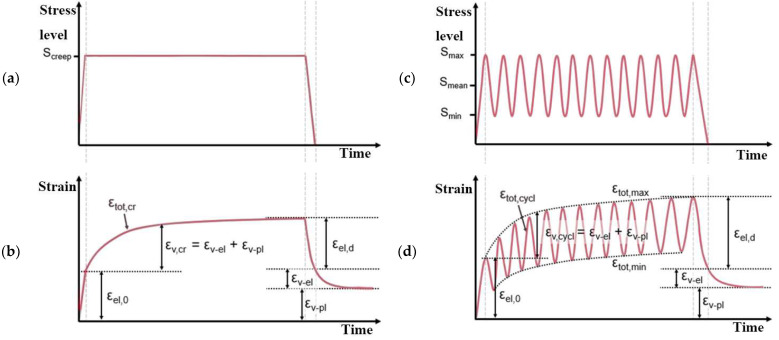
Schemes of static creep loading in time: (**a**) corresponding strain in time (**b**), cyclic creep loading in time (**c**), and corresponding strain development in time (**d**) (reproduced from [3]).

**Figure 2 materials-16-06128-f002:**
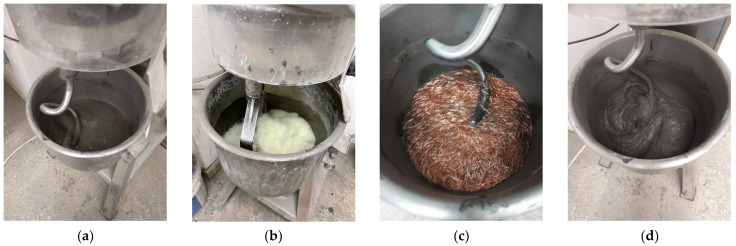
Plain (**a**) and fibre reinforced (**b**–**d**) specimen mix compositions in the mixing process [20].

**Figure 3 materials-16-06128-f003:**
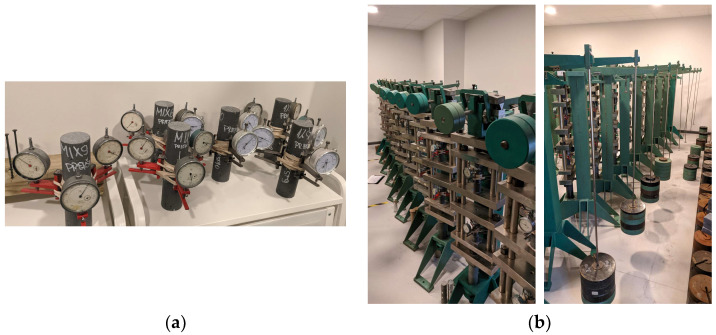
Plain and reinforced geopolymer specimens were used for shrinkage (**a**) and creep (**b**) testing.

**Figure 4 materials-16-06128-f004:**
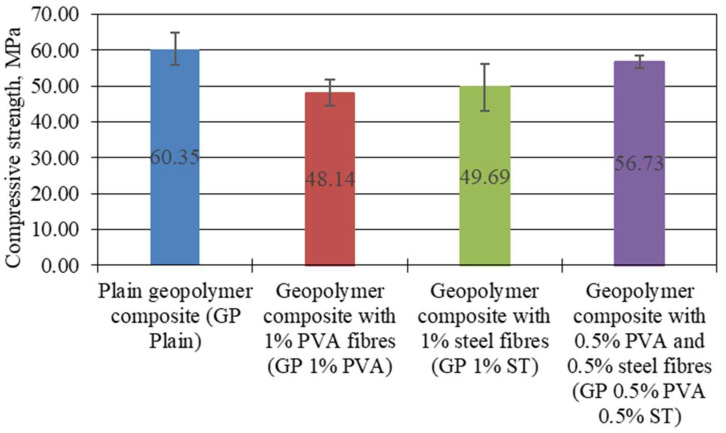
Geopolymer-composite compressive strength values prior to creep tests.

**Figure 5 materials-16-06128-f005:**
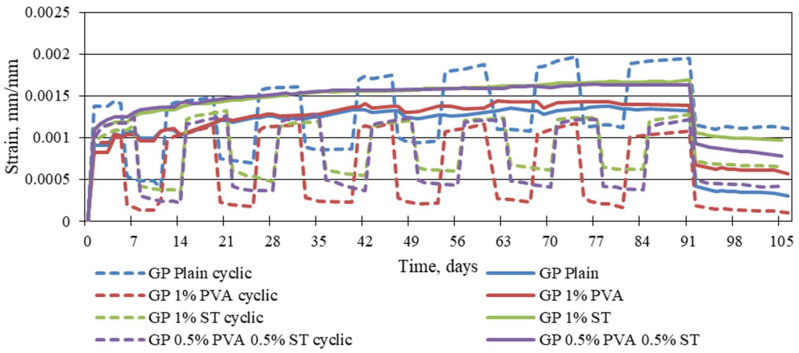
Reinforced and plain geopolymer-composite static and cyclic creep strains.

**Figure 6 materials-16-06128-f006:**
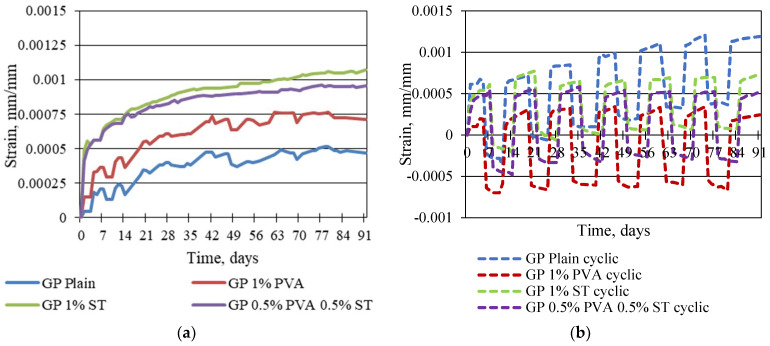
Reinforced and plain geopolymer composite static (**a**) and cyclic (**b**) creep strains without elastic strains.

**Figure 7 materials-16-06128-f007:**
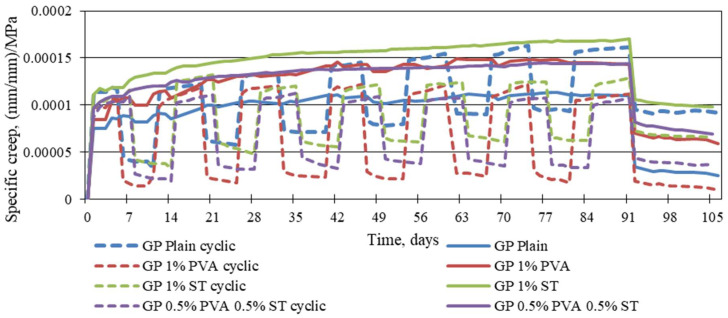
Reinforced and plain geopolymer-composite specific-static and specific-cyclic creep strains.

**Figure 8 materials-16-06128-f008:**
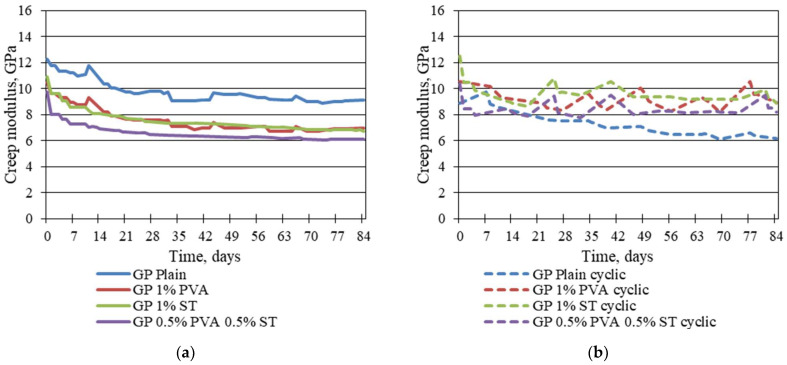
Modulus of creep development throughout creep tests for specimens tested on static creep (**a**) and cyclic creep (**b**).

**Figure 9 materials-16-06128-f009:**
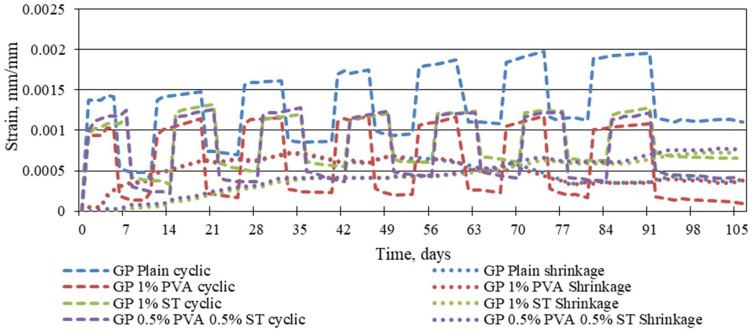
Cyclic creep strains and shrinkage strains of reinforced and plain geopolymer composites.

**Figure 10 materials-16-06128-f010:**
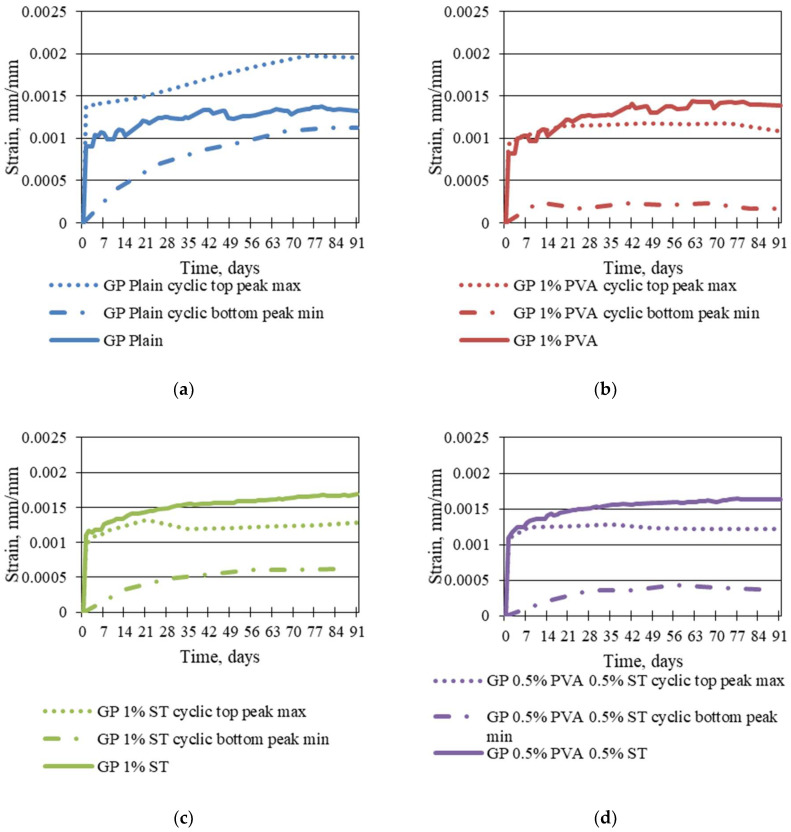
Plain (**a**) 1% PVA fibre-, (**b**) 1% steel fibre-, (**c**) and 0.5% PVA and 0.5% ST fibre (**d**)-reinforced geopolymer composite static creep and cyclic creep amplitude.

**Table 1 materials-16-06128-t001:** Fibre properties used in concrete composites [20,21].

Material	In-Use Diameter	Elongation (%)	Tensile Strength (MPa)	Density (g/cc)	Comment
Steel	0.25–1 mm	0.5–3.5	340–2800	7.6–7.85	High modulus of elasticity, low cost, widely available.
PVA	24–100 µm	6–12	1200–1600	1.1–1.3	Strong molecular bond, chemical resistance mainly to alkali environment.

**Table 2 materials-16-06128-t002:** Used geopolymer-composite alkali solution and dry mix quantitative parameters [20].

Dry Mix		Alkali Solution	
Ingredient	Weight Ratio	Ingredient	Weight
Quartz sand	1.00	NaOH flakes	400 g
Fly ash	1.00	Water	1000 g
Fibres	0.01	R-145 Na_2_O + SiO_2_ solution (molar module 2.5, density 1.45 g/cm^3^)	~3500 g

**Table 3 materials-16-06128-t003:** Chemical composition of the used fly ash from Skawina (Poland) coal power plant [23].

No.		%	No.		%
1.	Loss on ignition	2.84 ± 0.14	8.	Fe_2_O_3_	7.60
2.	SO_3_	0.95 ± 0.24	9.	SiO_2_ + Al_2_O_3_ + Fe_2_O_3_	78.21 ± 1.28
3.	Chloride (Cl)	0.034 ± 0.010	10.	MgO	3.06 ± 0.23
4.	CaO	0.02 ± 0.01	11.	P_2_O_5_	0.0008 (8 ± 1 mg/kg)
5.	SiO_2_ (reactive)	35.86 ± 0.64	12.	Na_2_O	1.72
6.	SiO_2_	47.81	13.	K_2_O	4.62
7.	Al_2_O_3_	22.80	14.	Na_2_O_eq_	4.76 ± 0.47

**Table 4 materials-16-06128-t004:** Basic properties of the used fibres [20].

Fibre Parameter	Steel Fibres (La Graminga GOLD)	PVA Mesofibres (Master Fiber 400/401)
Length	20.00 mm	18.00 mm
Diameter	0.30 mm	0.16 mm
Tensile strength	2635–3565 MPa	790–1160 MPa

**Table 5 materials-16-06128-t005:** Average values of standard deviation and coefficient of variation of static creep, cyclic creep, and shrinkage measurements.

		Plain GP	1% PVA	0.5% PVA/0.5% ST	1% ST
Static creep	Standard deviation	0.000089	0.0021	0.00012	0.00043
	Coefficient of variation	0.06668	0.15301	0.0681	0.27294
Cyclic creep	Standard deviation	0.00167	0.00029	0.00015	0.0006
	Coefficient of variation	0.1015	0.26915	0.13533	0.5815
Shrinkage	Standard deviation	0.00043	0.0000637	0.0000486	0.0002
	Coefficient of variation	0.10157	0.14463	0.16095	0.51044

## Data Availability

Not applicable.

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
