# Peer review of "Cyclic Load Impact Assessment of Long-Term Properties in Compression to Steel and Polyvinyl Alcohol Fibre-Reinforced Geopolymer Composites"

_materials, 2023, doi:10.3390/ma16186128_

Round 1

Reviewer 1 Report

The abstract should be re-written to better reflect on the originality and main findings. 

Lines 38-39 needs English improvement.

Overall, the introduction should be strengthened to better situate the context of the study, including the need to investigate this topic. In fact, the literature search on the geopolymer materials should be more comprehensive, and should refer to more documents/articles that are available. This topic has been well studied and documented, especially the use of fibers for this kind of applications. From the other end, the Fig 1 and the related discussion should be more succinct, since this is well known in the profession. Finally, the objectives of the study should be better highlighted at end of the introduction. 

The materials used are not well described, and also the composition of mixtures are not presented, which makes difficult the understanding of the paper. What was the chemical and physical properties of the fly ash? Please, provide more discussion about the fresh properties of mixtures considered. It would have been interesting to compare the geopolymer test results with cement-based mixtures? Why the authors selected 1% of fibers, please elaborate.

Why did the authors use a relative humidity of 30%? Please, elaborate more as this has direct effect on the results obtained. 

Lines 129-133: please, improve English. The discussion provided in this section is very hypothetic, and needs to be better discussed and supported by relevant literature. 

Fig. 3 to 6 are difficult to understand. The authors should better create some indices that would allow easier comparison of results, and draw conclusions accordingly. What were the flexural strengths of tested specimens, as these have direct effect on the test results. This section needs re-organization, while highlighting how many specimens were tested including their coefficients of variation. 

Overall, the idea is interesting and useful for the profession. In fact, the use of fibers in geopolymeric materials and their behavior to cyclic loading needs further investigations and understanding. Nevertheless, the paper is rather weak, and written in poor English. The paper needs major restructuring for the flow of ideas, and better description of the materials and methods used throughout.

English needs major improvements. 

Reviewer 2 Report

1. Line no. 18 and 19 are not clear

2. In materials and methods a photo of the specimen may be given so that the readers can understand the instrumentation of the specimen.

3. The Plain Geopolymer Concrete Compressive Strength is more than the fibre reinforced Geopolymer concrete. Hence the addition of fiber will not have any impact on compressive strength. Clarify.

4. In the Line No. 179,180 and 181, ' The specific creep value under static loading is surprisingly low for plain geopolymer composite and on average is 32.40% lower than the specific creep value under static load of 1% steel fiber reinforced geopolymer composite. Clarify.

5. In the conclusion the author may give one constructive suggestion keeping in mind all the work made in this paper.

OK

Reviewer 3 Report

An interesting article for researchers.

I think that small additions to the article are required.

According to the data presented in Figure 7. Why is the nature of the curves different for the following states: GP Plane and GP 1% ST, GP 1% PVA and GP 0.5% PVA 0.5% ST?

An English language test is required.

Reviewer 4 Report

Dear authors,

In this paper, the authors propose a study regarding the impact of cyclic load application on fly ash-based geopolymer composites reinforced with a low amount of fiber reinforcement, considering it as a weak element.

As a week elements

The abbreviation "PVA" in the title should be written out completely. The abstract needs to be expanded to have more than 200 words. Abbreviations should not be included in the abstract, and especially these need to be explained. For example, the same abbreviation "PVA."

The introduction should be augmented with more current references, thus increasing the number of references in the bibliography, which is currently quite concise. The Materials and Methods section should be supplemented with images and details of experimental trials (used recipes) and characteristics of the studied materials. Some microstructural investigations like SEM, TEM, etc., could also be introduced.

As a notable elements

The Results and Discussions chapter is well-written and appropriately accompanied by necessary graphs. The paper is interesting, but it necessitates the inclusion of the mentioned elements.

The editorial standards have not been met, and significant corrections and improvements are necessary for it to be reconsidered at this level. 

Round 2

Reviewer 1 Report

Overall, the introduction is written in weak English, and requires major restructuring and improvements. More information is needed regarding the effect of fibers and the comparison between steel and PVA fibers, especially that this topic has been very well documented in the literature. The paper’s objectives and novelty should be better reflected at the end of introduction. 

Lines 100-107 are not well described. It is very important to better elaborate on the GP mix design, and the fibers properties (the reference cannot be enough). This comment was raised in the previous submission.

The results and discussion section should be divided into sub-sections to better convey the ideas and simplify the reading. 

The discussion provided throughout lines 142-147 is not convincing, since there are many other references in the literature that showed different tendencies!! Also, the lines 157-161 are very hypothetical, since the authors cannot transpose the context and results of another study to their current one. Pls, it is important to give the size and other properties of the used fibers (otherwise, the discussion cannot be applicable); the same idea applies when discussing the shrinkage and creep results. 

The discussion of the continuing part of the paper is not well written or interpreted, and full of English mistakes and grammatical errors. Lines 231-247 are not understood, or well explained. 

The reviewer strongly believes that the paper requires major English editing and improvements by an English native scientist, since it is full with grammatical and sentence errors, making difficult the understanding of the idea discussed. It is not appropriate to MDPI readers! 

The reviewer strongly believes that the paper requires major English editing and improvements by an English native scientist, since it is full with grammatical and sentence errors, making difficult the understanding of the idea discussed. It is not appropriate to MDPI readers! 

Reviewer 4 Report

Dear authors,

Firstly thank you for your feedback!

Even if not completely (see abstract and bibliography) the authors have responded to my comments, the scientific quality of the paper has been greatly improved according to all reviewers’ observations and that is better approaching of the journal exigencies. 

Author Response

Thank you for the feedback. The abstract, introduction, results & discussion sections have been slightly improved.